# Obesity, Burden of Ischemic Heart Diseases and Their Ecological Association: The Case of Uzbekistan

**DOI:** 10.3390/ijerph191610447

**Published:** 2022-08-22

**Authors:** Murodkhon Marufkhonovich Usmanov, Odgerel Chimed-Ochir, Bilegt Batkhorol, Yui Yumiya, Lola Mamazairovna Hujamberdieva, Tatsuhiko Kubo

**Affiliations:** 1Department of Public Health and Health Policy, Graduate School of Biomedical and Health Sciences, Hiroshima University, Hiroshima 734-8553, Japan; 2Department of Epidemiology and Biostatistics, School of Public Health, Mongolian National University of Medical Sciences, Ulaanbaatar 14210, Mongolia

**Keywords:** obesity, body mass index, the burden of ischemic heart disease, ecological association, Uzbekistan

## Abstract

Ischemic heart diseases are the leading cause of death in Uzbekistan. There are numerous risk factors affecting ischemic heart disease, and obesity is one of the major independent risk factors. This study is the first attempt to estimate the ecological association between obesity prevalence and the burden of ischemic heart disease between 1990 and 2019 in Uzbekistan. To define the prevalence of all obesity types, death, and incidences of ischemic heart disease for certain periods, the Joinpoint regression tool was used. A separate linear regression analysis was performed to analyze the relationship between obesity and ischemic heart disease mortality and morbidity. A positive linear relation was found between the prevalence of obesity types and incidence/death rates for both sexes (r = 0.59–0.87). All types of obesity were highly significant positive predictors of incidence of and death from ischemic heart disease (*p* < 0.0001). The slope (B1) suggested that for an increment in obesity prevalence of 1% among adults aged over 20, the incidence of ischemic heart disease increased by 40.2 (*p* < 0.0001) and 38.3 (*p* < 0.0001) per 100,000 persons for men and women, respectively. The current country-level conclusions are valuable, because it allows decision makers to draw specific conclusions, applicable at the state and local level for policymaking.

## 1. Introduction

The Republic of Uzbekistan (Uzbekistan) is the most populous country in Central Asia, with a population of 32,768,725 in 2019, of which 28.2% account for its youth [1]. The health care system is fully financed by general government revenue, and overall health is managed by the Ministry of Health of Uzbekistan, with subdivisions of the Ministry in each region [2]. Government budget and private resources, mainly in the form of out-of-pocket expenditure, are the main revenue sources for health expenditure, and voluntary health insurance does not play a major role [2].

Circulatory system diseases are the leading cause of death in Uzbekistan. Ischemic heart disease (IHD) is an atherosclerotic disease that is inflammatory in nature [3], characterized by persistent or unstable angina, myocardial infarction, or sudden cardiac death [4], and is becoming a major cause of death in both developing and developed countries.

Between 1990 and 2017, the death rate from IHD increased by 77.2% in Uzbekistan, which was the highest increase in the world [5]. There are numerous risk factors affecting IHD, with obesity being one of the major independent risk factors, as demonstrated by many cohort studies [6,7]. Obesity, which is defined as an excessive excess accumulation of fatty tissue that increases the risk of diseases such as diabetes, hypertension, and cardiovascular disease, is one of primary care’s main tasks [8]. Obesity has a strong link to coronary artery disease. Patients with a greater body mass index had more advanced cardiovascular problems than those with a normal body mass index [9].

A great number of attempts have been made to evaluate the association between obesity and cardiovascular diseases, and a wealth of clinical and epidemiological evidence exists from cohort studies conducted all over the world. However, there is a lack of country-level scientific evidence on non-communicable diseases in Uzbekistan. For the period between 1945 and March 2022, a total of 1966 articles can be found in PubMed, of which 164 are related to non-communicable diseases, and no research was conducted in terms of the association between obesity and IHD. Thus, the current study is the first attempt to estimate the ecological association between obesity prevalence and the burden of IHD in Uzbekistan. We believe that country-level conclusions are valuable, despite a great amount of available evidence from all over the world, because it allows decision makers to draw specific conclusions, applicable at the state and local level for policymaking.

Therefore, we aimed to evaluate trend of obesity prevalence, burden of ischemic heart disease and their ecological association in Uzbekistan, from 1990 to 2019.

## 2. Materials and Methods

### 2.1. Data Source

For obesity data, we extracted data from the Non-Communicable Diseases Risk Factor Collaboration (NCD-RisC) [10], a network of health scientists from around the world that provides data on major risk factors for non-communicable diseases for all of the world’s countries, regarding prevalence (% of the total population aged over 20) of obesity between 1987 and 2016 in Uzbekistan. The extracted data included the prevalence of obesity (BMI ≥ 30), Class I obesity (30 < BMI ≤ 35), Class II obesity (35 < BMI ≤ 40), and Class III obesity (BM > 40) in both men and women. The body mass index was used as a measurement of obesity, because it is a simple index of weight-for-height that is commonly used to classify overweight and obesity in adults [11]. Although data on obesity prevalence are available, because data on IHD are only available for the period between 1990 and 2019, we extracted data on obesity prevalence between 1987 and 2016 by reflecting the 3-year lag periods between exposure and outcome used in the WHO Multinational Monitoring of Trends and Determinants in Cardiovascular Disease (MONICA) Project [12].

For the IHD data, we downloaded the data from the Global Health Data Exchange of the Institute of Health Metrics and Evaluation, the University of Washington USA regarding sex-specific incidences and deaths of IHD, and their age-standardized rates (per 100,000 population) between 1990 and 2019 in Uzbekistan [13], to determine the trend of the burden of ischemic heart disease. The diseases associated with ischemic heart disease that we studied were I20-I21.6, I21.9-I25.9, Z82.4-Z82.49, as classified by the International Classification of Diseases, 10th revision (ICD-10).

### 2.2. Data Analysis

To describe the prevalence of all obesity types, number- and age-standardized rates of death and incidences of ischemic heart disease for selected periods, the Joinpoint regression tool, version 4.9.1.0—22 April (Statistical Methodology and Applications Branch, Surveillance Research Program, National Cancer Institute, Bethesda, MD, USA) was used to identify the average annual percent changes between these years. The equation of the regression model is as follows:log (*R_y_*) = *b*_0_ + *b*_1_*y*
where log (*R_y_*) is the natural log of the rate in year *y*. The average percentage change from year *y* to year *y* + 1 was calculated as shown below.
$$\lfloor \frac{R_{y+1}-R_{y}}{R_{y}}\rfloor \times 100=\frac{\left\{e^{b_{0}+b_{1}\left(y+1\right)-}e^{b_{0}+b_{1}\left(y\right)}\right\}}{e^{b_{0}+b_{1}\left(y\right)}}\times 100=\left(e^{b_{1}}-1\right)\times 100$$

The average annual percentage changes (AAPC) over the periods 1990–2000, 2000–2010, 2010–2019, and 1990–2019 were calculated using a weighted average of the slope coefficients of the underlying joinpoint regression line with the weights equal to the length of each segment over the intervals. If we denote *b_i_* as the slope coefficients for each segment in the desired range of years, and the *w_i_* as the length of each segment in the range of years, then AAPC would be $\left\{Exp\left(\frac{\sum^{\;}w_{i}b_{i}}{\sum^{\;}w_{i}}\right)-1\right\}\times 100$. Joinpoint tests of significance use a Monte Carlo permutation method to determine the statistically significant changes across successive study periods.

Linear regression analysis was performed for each sex, with age-adjusted incidence and death rates of ischemic heart disease as a dependent variable and prevalence of each type of obesity as the independent variables. In each linear regression analysis, 30 pair observations of the independent variable and dependent variable (30 years between 1990 and 2019) were used. As IHD is a condition that can be caused by multiple factors, we considered including other confounding factors such as high blood pressure, physical inactivity, smoking, and low-density lipoprotein in the regression analysis. However, these variables were strongly correlated with each other, which may lead to a multicollinearity problem. The Variance Inflating Factor (VIF) was then calculated on the multivariate regression model (age-adjusted incidence and death rates of IHD as a dependent variable and prevalence of obesity, high blood pressure, smoking, and mean low-density lipoprotein as independent variables) to see if the collinearity of these variables may affect the model utilizing those variables to predict IHD. The generated VIFs ranged from 114 to 399, indicating significant collinearity that affected the regression model. Multicollinearity reduces the precision of the estimated coefficients, which weakens the statistical power of the regression model [14]. In addition, all of these confounding variables, including high blood pressure, low-density lipoprotein and smoking, were on a casual pathway between obesity and IHD. This is called an over-adjustment, and if an over-adjustment problem occurs, the total effect of obesity on IHD cannot be reliably estimated [15]. Therefore, we conducted a univariate analysis with obesity as an independent variable and IHD as a dependent variable. Microsoft Excel (Microsoft Corp.; Redmond, Washington DC, USA) and STATA 15 (STATA Corp; College Station, TX, USA) were used for analysis.

## 3. Results

Table 1 shows the prevalence of obesity in Uzbekistan between 1987 and 2016, and the average annual percentage changes. As of 2016, 14.37% of men and 19.86% of women had obesity (BMI > 30). In terms of obesity type, 11.52% of men and 12.91% of women had Class I obesity, 2.24% of men and 4.88% of women had Class II obesity, and 0.6% of men and 2.08% of women had Class III obesity. Between 1987 and 2016, the prevalence of overall obesity and all types of obesity consistently increased for both men and women, with average annual percentage changes of 3.57% and 2.89%, respectively (*p* < 0.0001). Class III obesity among men (8.60%) and women (6.42%) had the highest average annual percentage change between 1987 and 2016.

Throughout the study period, for both men and women there was a steady increase in the prevalence of all types of obesity, with women experiencing a higher prevalence than men (Figure 1).

The burden of ischemic heart disease in Uzbekistan between 1990 and 2019 is shown in Table 2. In 2019, the age-standardized incidence rate for men was 1178.4 per 100,000 population (N = 84.349) and 873.9 for women (N = 59,386), respectively. The number of deaths due to ischemic heart disease was 40,615 for men and 34,886 for women, which accounted for 36.1% and 38.4% of all-cause death, respectively.

The age-standardized death rate was 792.7 and 640.8 per 100,000 population for men and women, respectively. During the study period, the age-standardized incidence rate for both men and women increased annually by 1.39% and 2.24%, respectively (*p* < 0.0001). The age-standardized death rate for both men and women increased annually by 2.41% and 3.25%, respectively (*p* < 0.0001).

Figure 2 shows the trend of death and incidence rate during the study period. Both the age-standardized incidence and death rates increased annually between 1990 and 2010, and an annual decrease was observed between 2010 and 2019.

The scatter plots shown in Figure 3 and Figure 4 depict a positive linear relationship between the prevalence of obesity types and incidence/death rates for both men and women (r = 0.59–0.87).

All types of obesity were highly significant positive predictors of incidence of and death from ischemic heart disease (Table 3), with an adjusted R2 ranging from 0.35 to 0.81 (*p* < 0.0001). The slope (B1) suggests that, for each increment in obesity prevalence of 1% among adults aged over 20, the incidence of ischemic heart disease increased by 40.2 (95%CI = 24.9–55.3; R2 = 0.51; *p* < 0.0001) and 38.3 (95%CI = 30.3–46.3; R2 = 0.78; *p* < 0.0001) per 100,000 persons for men and women, respectively. Similarly, the deaths of men and women were increased by 44.6 (95%CI = 27.4–61.7; R2 = 0.50; *p* < 0.0001) and 40.5 (R2 = 0.71; *p* < 0.0001) per 100,000 persons, respectively. For both men and women, the slopes of regression tend to increase as the severity of obesity increases.

## 4. Discussion

In the current study, we investigated the ecological relationship between obesity and the burden of ischemic heart disease between 1990 and 2019 in Uzbekistan. In both men and women, significantly positive ecological relations between all types of obesity and incidence of and death from ischemic heart disease were recorded. In both men and women, age-standardized incidence and death rates increased by between 38.3 and 44.6 per 100,000 persons per obesity prevalence of 1% among adults. Age-adjusted incidence and death rates also consistently increased with crease increasing severity of obesity. The increase in incidences and deaths per incremental increase in obesity prevalence was higher in men than in women.

An ecological study showing a relationship between obesity and heart disease might be less important, because there is a wealth of clinical and epidemiological evidence linking obesity to cardiovascular diseases that has been derived from cohort studies [16,17,18]. Although this ecological relationship lacks the strengths of formal exposure–response analyses, its public health importance should not be underestimated. Despite the numerous findings that have been published on the basis of observational studies in developed countries, ecological studies are still being conducted to investigate the association between obesity and cardiovascular diseases in these countries [19,20]. Therefore, however, given that the current study is the first attempt to estimate the association between obesity prevalence and burden of ischemic heart disease in Uzbekistan, we believe that country-level conclusions are valuable, as they can inform and suggest important policy actions to benefit public health.

As seen in the regression analysis, increased burden of ischemic heart disease was linearly associated with severity of obesity, which may reflect the fact that the prevention of larger proportions of the population experiencing severe obesity may have a great impact on reducing the burden of ischemic heart disease in Uzbekistan. We also observed consistent increases in obesity prevalence, with much higher annual percentage increases in the prevalence of Class III obesity during the study period in Uzbekistan. Figure 1 also shows a consistent increase in obesity prevalence in Uzbekistan between 1987 and 2016. One of the possible factors contributing to obesity is physical inactivity [21]. Engagement in physical activity is affected by many factors, including economics, social norms, cultural backgrounds, demographics, and environment [22]. Uzbekistan has a more service-based economy, and people working in the service sector have a more sedentary lifestyle than those working in either agriculture or industry [23]. In addition to sedentary lifestyle, there is an increasing number of people who drive a car in their daily life, and spend more time spent surfing the Internet and watching television than engaging in outdoor activities, contributing to physical inactivity in Uzbekistan. In addition, the traditional food culture is full of fat-laden foods, such as the national dish “plov”, which is made from fatty meat and white rice and cooked in a rich amount of animal fat or vegetable oil [24]. Each meal is also complemented by a huge portion of bread. Although the etiology of obesity is highly complex and includes dietary, physiological, genetic, psychological, environmental, social, and economic components [25], in Uzbekistan, the abovementioned factors are likely to have contributed to the increase in obesity prevalence. Therefore, there is a need for future research on factors of obesity and the development of highly effective interventions that aim to control these direct and indirect factors through well-targeted government policies, as well as health education and promotion programs. According to our findings, one in every three deaths in 2019 occurred because of ischemic heart disease in both men and women. Thus, reducing mortality due to ischemic heart disease by preventing obesity may even contribute significantly to decreasing all-cause mortality. Policymakers may need to improve early detection systems for ischemic heart disease among populations with obesity. Both individual and organizational efforts can aide in successful early detection. As individuals, Uzbeks do not tend to seek health care until complications arise. In some rural areas, primary- and secondary-level hospitals are not fully equipped with the necessary medical equipment, or much of the available equipment is outdated. As a result, necessary diagnostic tests, such as electrocardiogram, echocardiography, cardiac stress test and coronary computed tomography are unavailable. Patients who need detailed medical examinations must travel to other tertiary-level public or private hospitals, and disadvantaged patients cannot afford the travel and medical fees. As a result, people only receive medical care in the case of life-threatening emergencies [26]. Since the late 1990s, the Uzbek Government has been building its commitment to implementing activities to prevent and control NCDs. In 2003, the government of Uzbekistan implemented the “Year of Health”. The aim was to evaluate the health care system [27]. Also in 2003, the national projects “Health-1” and “Health-2” were launched to improve public health infrastructure and functions and primary health care services, and to train medical professionals [28]. However, formal assessment of the short- and long-term outcomes of these programs is ultimately necessary to inform further management.

The current study has strengths and limitations, and lends itself to future considerations. The most noteworthy strength is that, as mentioned previously, this is the first attempt to estimate the association between the prevalence of obesity and the burden of ischemic heart disease in Uzbekistan. We believe the study findings will drive the attention of policymakers toward the development of beneficial countermeasures. Our study presents some limitations in study design, and suggests some future considerations. First, the analyses performed in the current study were based on data aggregated for populations. It is impossible to draw conclusions regarding individual-level associations, because the use of aggregated data means that information on the joint distribution within groups is lacking [29]. To completely avoid this potential ecological bias, individual-level data are necessary; thus, future observational research at the patient level is encouraged. Second, the magnitude of ecological effects depends not only on the biological effect of obesity, but also on contextual effects such as degree of access to and pattern of health care services and other aspects, but we were not able to adjust these factors in the model. Third, the independent variable used as a measurement of obesity was BMI, calculated as body weight in kilograms divided by the height in meters squared (kg/m^2^). There is debate as whether BMI gives a precise idea of body composition with respect to effects on health risks related to excess weight consisting of fat or fat distribution. Other methods including waist circumference and central and peripheral fat mass have also been used to define obesity. Nevertheless, BMI is an internationally accepted standard method used by researchers and others dealing with human health, despite the existence of alternatives [30]. Fourth, national data are not publicly available in Uzbekistan, and the national data reported to the WHO cover only 12 inconsecutive years. Therefore, we used the data estimated by the Institute of Health Metrics and Evaluation, University of Washington, USA. Thus, another potential source of bias is that the data on IHD are estimated data, and may not be fully accurate as national data. Therefore, health policymakers and the relevant authorities in Uzbekistan should address both the unavailability of data and the quality of the reported data, and should be aware of the importance of making national data available for research purposes.

Although we discuss the obesity prevalence and burden of ischemic diseases in Uzbekistan at the national level herein, there is a huge disparity between rural and urban areas in terms of people’s lifestyles, infrastructure, and the availability of and access to health service. Overall, these patterns have a substantial diversity, and the status of health in Uzbekistan would be better described at the subnational level.

## 5. Conclusions

Ischemic heart disease is currently a major cause of mortality among all causes of mortality in Uzbekistan. Additionally, the incidence of obesity has increased in the last three decades in Uzbekistan. In Uzbekistan, significant ecologic associations were observed between prevalence of all types of obesity and age-adjusted incidence and death rates from ischemic heart disease for both men and women between 1990 and 2019. The current country-level conclusions are valuable, despite the great amount of available evidence from all over the world, because they allow decision makers to draw specific conclusions, applicable at the state and local level for policymaking.

## Figures and Tables

**Figure 1 ijerph-19-10447-f001:**
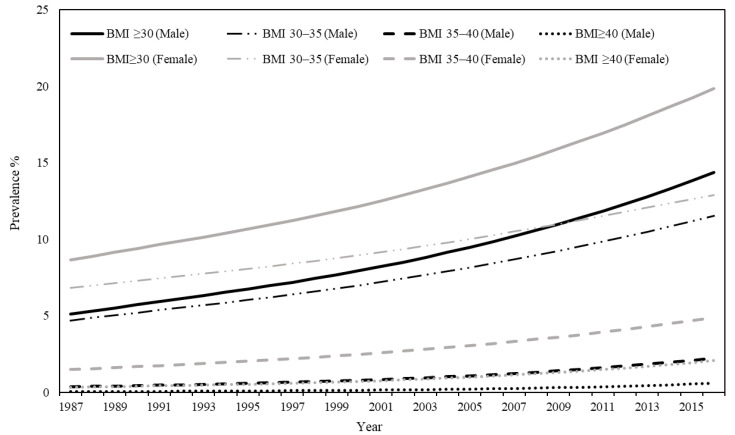
Trend of obesity between 1990 and 2016 in Uzbekistan.

**Figure 2 ijerph-19-10447-f002:**
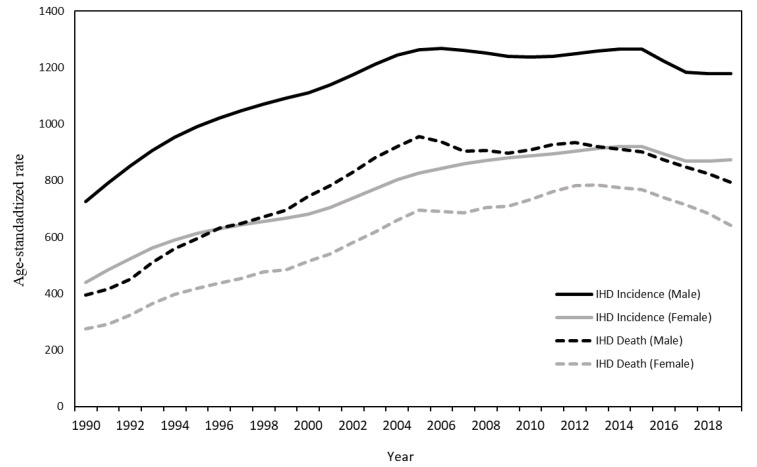
The trend of the age-standardized rate of death from and incidence of ischemic heart disease between 1990 and 2019 in Uzbekistan.

**Figure 3 ijerph-19-10447-f003:**
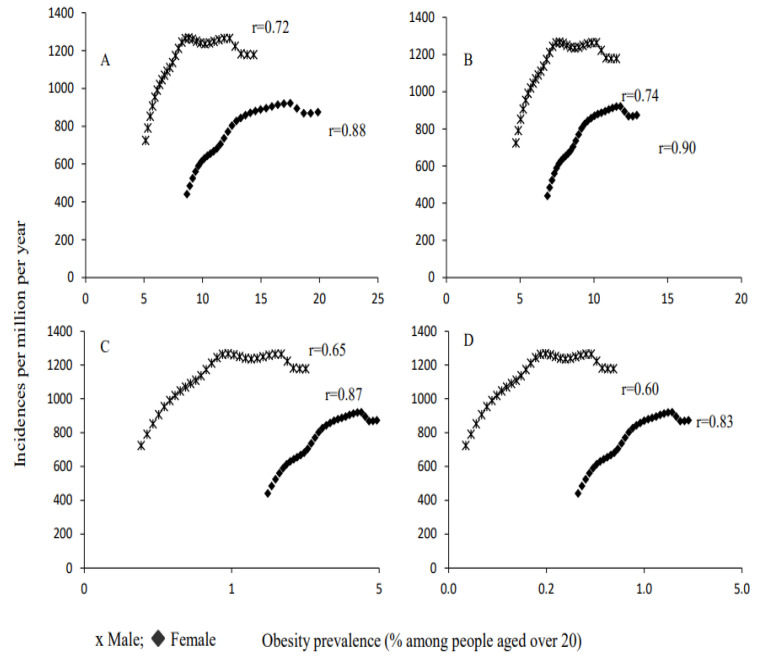
Ecological relationships between incidence rates of ischemic heart disease and obesity prevalence. (**A**) Obesity (BMI > 30); (**B**) Class I (BMI = 30–34.9); (**C**) Class II (BMI = 35–39.9); (**D**) Class III (BMI ≥ 40).

**Figure 4 ijerph-19-10447-f004:**
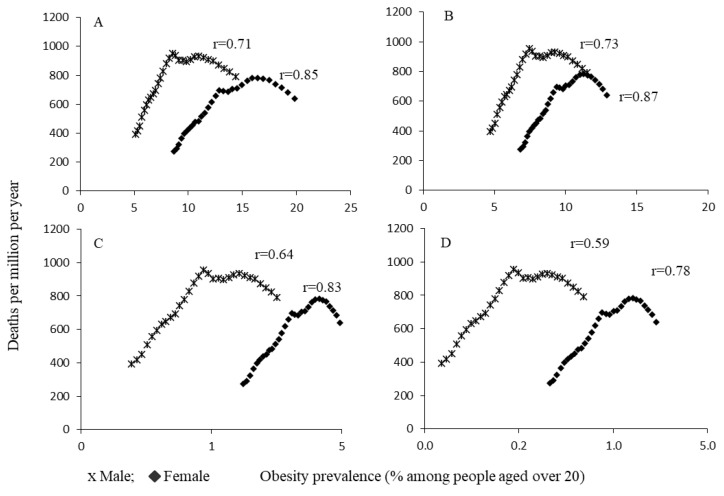
Ecological relationships between death rates for ischemic heart disease and obesity prevalence. (**A**) Obesity (BMI > 30); (**B**) Class I (BMI = 30–34.9); (**C**) Class II (BMI = 35–39.9); (**D**) Class III (BMI ≥ 40).

**Table 1 ijerph-19-10447-t001:** Prevalence of obesity and its average annual percentage change in Uzbekistan between 1990 and 2019.

Class of Obesity	Prevalence ^1^				Average Annual Percentage Change (%) ^2^			
	1987	1997	2007	2016	1987–1997	1997–2007	2007–2016	1987–2016
**Men**								
Obesity (BMI ≥ 30)	5.13	7.20	10.21	14.37	3.39	3.46	3.89	3.57
Class I (BMI 30.0–34.9)	4.71	6.40	8.69	11.52	3.07	3.05	3.22	3.11
Class II (BMI 35.0–35.9)	0.37	0.68	1.24	2.24	6.07	6.09	6.88	6.30
Class III (BMI ≥ 40)	0.05	0.12	0.27	0.60	8.61	8.30	9.26	8.60
**Women**								
Obesity (BMI ≥ 30)	8.67	11.23	14.97	19.86	2.56	2.91	3.23	2.89
Class I (BMI 30.0–34.9)	6.85	8.41	10.51	12.91	2.03	2.25	2.35	2.21
Class II (BMI 35.0–35.9)	1.48	2.20	3.32	4.88	3.97	4.18	4.40	4.17
Class III (BMI ≥ 40)	0.34	0.61	1.14	2.08	5.99	6.41	6.96	6.42

^1^ Percentage (%) of the total population. ^2^ Please see the methods for computation of Average Annual Percentage Change in Methods section. All average annual percentage changes were statistically significant.

**Table 2 ijerph-19-10447-t002:** Death and incidence of ischemic heart disease in Uzbekistan.

	Number						% of All Cause ^1^			ASR		Average AnnualPercentage Change (%) ^2^				
	1990	2000	2010	2019	1990	2000	2010	2019	1990	2000	2010	2019	1990–2000	2000–2010	2010–2019	1990–2019
Incidence																
Male	32,152	49,648	66,649	84,349	0.08	0.10	0.12	0.13	724.6	1110.5	1236.0	1178.4	4.42	1.20	−0.76	1.39
Female	27,655	42,449	49,423	59,386	0.06	0.08	0.08	0.08	440.4	681.9	887.7	873.9	4.28	2.58	−0.48 *	2.24
Death																
Male	15,488	27,603	37,405	40,615	22.2	28.6	36.0	36.1	394.7	744.3	909.5	792.7	7.00	1.93	−1.83	2.41
Female	16,611	29,844	34,749	34,886	27.2	34.3	39.5	38.4	274.8	514.3	733.5	640.8	6.75	3.86	−1.44	3.25

^1^ Total number of all-cause deaths can be found at http://ghdx.healthdata.org/gbd-results-tool; ^2^ the method for the computation of the Average Annual Percentage Change can be found in the Materials and Methods section; All average annual percentage changes are statistically significant unless indicated with *.

**Table 3 ijerph-19-10447-t003:** Regression analysis on age-standardized incidence (A) and death rates (B) of ischemic heart disease versus prevalence of obesity between 1990 and 2019 in Uzbekistan.

A								
Class of Obesity	Regression Parameters						Adjusted R^2^	*p* Value
	B_0_	(95% CI)	*p* Value	B_1_	(95% CI)	*p* Value		
**Male**								
Obesity (BMI ≥ 30)	772.9	(632.1–913.6)	<0.0001	40.2	(24.9–55.3)	<0.0001	0.51	<0.0001
Class I (BMI 30.0–34.9)	704.3	(549.6–859.0)	<0.0001	55.8	(36.1–75.4)	<0.0001	0.55	<0.0001
Class II (BMI 35.0–35.9)	942	(845.9–1037.8)	<0.0001	182.3	(99.7–264.9)	<0.0001	0.42	0.0001
Class III (BMI ≥ 40)	996.7	(914.6–1079.1)	<0.0001	587.3	(287.4–887.3)	<0.0001	0.36	0.0004
**Female**								
Obesity (BMI ≥ 30)	248.7	(139.6–357.9)	<0.0001	38.3	(30.3–46.3)	<0.0001	0.78	<0.0001
Class I (BMI 30.0–34.9)	76.9	(−53.0–206.9)	0.236	71.6	(58.2–85.1)	<0.0001	0.81	<0.0001
Class II (BMI 35.0–35.9)	403	(320.5–485.5)	<0.0001	125.0	(97.6–152.5)	<0.0001	0.76	<0.0001
Class III (BMI ≥ 40)	535.5	(468.8–602.2)	<0.0001	235.3	(172.9–297.6)	<0.0001	0.68	<0.0001
**B**								
**Class of Obesity**	**Regression parameters**						**Adjusted R^2^**	***p* value**
	**B_0_**	**(95% CI)**	***p* value**	**B_1_**	**(95% CI)**	***p* value**		
**Male**								
Obesity (BMI ≥ 30)	376.6	(217.8–535.4)	<0.0001	44.6	(27.4–61.7)	<0.0001	0.50	<0.0001
Class I (BMI 30.0–34.9)	299.9	(125.4–474.3)	0.001	61.9	(39.8–84.1)	<0.0001	0.54	<0.0001
Class II (BMI 35.0–35.9)	564.8	(456.6–672.9)	<0.0001	201.6	(108.5–294.7)	<0.0001	0.41	<0.0001
Class III (BMI ≥ 40)	626.1	(533.3–718.9)	<0.0001	647.2	(308.7–985.6)	<0.0001	0.35	0.0005
**Female**								
Obesity (BMI ≥ 30)	51.2	(−84.7–187.1)	0.447	40.5	(30.6–50.5)	<0.0001	0.71	<0.0001
Class I (BMI 30.0–34.9)	−133.5	(−297.4–30.5)	0.107	76.1	(59.1–93.0)	<0.0001	0.75	<0.0001
Class II (BMI 35.0–35.9)	215.2	(113.2–317.3)	<0.0001	132.1	(98.1–166.0)	<0.0001	0.69	<0.0001
Class III (BMI ≥ 40)	357.2	(276.2–438.2)	<0.0001	246.2	(170.5–321.9)	<0.0001	0.61	<0.0001

B_0_ = intercept of regression line. B_1_ = slope of regression line. Regression model: Age-standardized mortality/incidence rates of ischemic heart disease (per hundred population) = B_0_ + B_1_ × prevalence of obesity (% of total population).

## Data Availability

The data presented in this study: The data of Obesity available from Non-Communicable Diseases Risk Factor Collaboration [https://ncdrisc.org/ (accessed on 14 September 2021)], the data of burden of ischemic heart disease available from Global Health Data Exchange of the Institute of Health Metrics and Evaluation [https://ghdx.healthdata.org/ (accessed on 24 September 2021)], the data of population available from United Nations Department of Economic and Social Affairs Population Dynamics [https://population.un.org/wpp/ (accessed on 21 September 2021)].

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
