# Peer review of "Obesity, Burden of Ischemic Heart Diseases and Their Ecological Association: The Case of Uzbekistan"

_ijerph, 2022, doi:10.3390/ijerph191610447_

Round 1

Reviewer 1 Report

The authors have written a manuscript based on generic data that have limited validity, as indicated in the discussion. In addition, the information sent to the journal, or received by the reviewers, is incomplete, so the article requires a major revision in order to be accepted in the International Journal of Environmental Research and Public Health.

Major revisions necessary.

1.- The authors do not have individual data, and furthermore the data is incomplete, as indicated in the discussion.

2.-IHD is a multifactorial disease, and the authors study only BMI, and other important parameters such as blood pressure appear, which appear on the web page used by the authors (attached file). The authors need to study all available variables related to IHD.

3.- The data available and the conclusions obtained are discreet, and are not up to the standards of the journal.

4.- Between lines 185 and 189, the authors need to indicate the reference where other similar studies appear as indicated.

5.- Following from line 189, the authors indicate that it is the first study done in Uzbekistan. Why not compare it with data from one or more other studies in other countries? Lines 202 and 203 also indicate that the conditions with other countries may be different, but they do not clearly indicate the differences or characteristics of the country, aand diferences with others.

6.- In line 198 the authors refer to a supplementary figure 2, but I have not received any supplementary figure, so the supplementary material should be included.

7.- On lines 212-216, the authors indicate the minimum information required for an article of these characteristics, and with individual data to be able to carry out correlation studies, as indicated between lines 242 to 245, and they also indicate on line 263 that IHD data are estimated.

8.- At the end of the discussion in lines 266 to 269, the authors talk about the differences between rural and urban populations. With a good design of the database, these two groups should be able to be differentiated, and thus be able to study, obtain tables or graphs with that difference, ...

For all these major review points, the article must be reanalyzed, completing the database, and rewritten in order to be accepted.

Author Response

Thank you very much for your valuable time in reviewing our manuscript.  We have attached our point-by-point response. 

Reviewer 2 Report

This study is the first attempt to estimate the ecological association between obesity prevalence and the burden of ischemic heart diseases in Uzbekistan. A very last advice is to recommend the authors to use the term “people with obesity” or “population with obesity” as advised by the latest Guidelines to manage obesity instead of the term “obese” or overweight that may not currently sound politically correct. It is important to make those modifications in the article.

Author Response

Thank you for your valuable time in reviewing our manuscript.  Thank you very much for your advice. We reflected your comments and corrected “obese people” as “population with obesity”.

Round 2

Reviewer 1 Report

Congratulations to the authors for this new version, which corrects the deficiencies of the previous version and should be accepted by IJERPH.